# AGEs and sRAGE Variations at Different Timepoints in Patients with Chronic Kidney Disease

**DOI:** 10.3390/antiox10121994

**Published:** 2021-12-15

**Authors:** Paolo Molinari, Lara Caldiroli, Elena Dozio, Roberta Rigolini, Paola Giubbilini, Massimiliano M. Corsi Romanelli, Piergiorgio Messa, Simone Vettoretti

**Affiliations:** 1Unit of Nephrology, Dialysis and Kidney Transplantation, Fondazione IRCCS Ca’ Granda Ospedale Maggiore Policlinico di Milano, 20122 Milan, Italy; paolo.molinari1@unimi.it (P.M.); lara.caldiroli@policlinico.mi.it (L.C.); piergiorgio.messa@gmail.com (P.M.); 2Laboratory of Clinical Pathology, Department of Biomedical Science for Health, Università degli Studi di Milano, 20133 Milan, Italy; elena.dozio@unimi.it (E.D.); mmcorsi@unimi.it (M.M.C.R.); 3Service of Laboratory Medicine1-Clinical Pathology, IRCCS Policlinico San Donato, San Donato Milanese, 20097 Milan, Italy; roberta.rigolini@grupposandonato.it (R.R.); paola.giubbilini@grupposandonato.it (P.G.); 4Department of Clinical Sciences and Community Health, Università degli Studi di Milano, 20122 Milan, Italy

**Keywords:** advanced glycation end-products (AGE), chronic kidney disease (CKD), soluble receptor for AGE (sRAGE), cleaved RAGE (cRAGE), endogenous secretory RAGE (esRAGE)

## Abstract

Patients with chronic kidney disease (CKD) are affected by enhanced oxidative stress and chronic inflammation, and these factors may contribute to increase advanced glycation end-products (AGEs). In this study we quantified AGEs and soluble receptors for AGE (sRAGE) isoforms and evaluated the association between their variations and eGFR at baseline and after 12 months. We evaluated 64 patients. AGEs were quantified by fluorescence intensity using a fluorescence spectrophotometer, and sRAGE by ELISA. Median age was 81 years, male patients accounted for 70%, 63% were diabetic, and eGFR was 27 ± 10 mL/min/1.73 m^2^. At follow up, sRAGE isoforms underwent a significant decrement (1679 [1393;2038] vs. 1442 [1117;2102], *p* < 0.0001), while AGEs/sRAGE ratios were increased (1.77 ± 0.92 vs. 2.24 ± 1.34, *p* = 0.004). Although AGEs and AGEs/sRAGE ratios were inversely related with eGFR, their basal values as well their variations did not show a significant association with eGFR changes. In a cohort of patients with a stable clinical condition at 1 year follow-up, AGEs/sRAGE was associated with renal function. The lack of association with eGFR suggests that other factors can influence its increase. In conclusion, AGEs/sRAGE can be an additional risk factor for CKD progression over a longer time, but its role as a prognostic tool needs further investigation.

## 1. Introduction

Patients with chronic kidney disease (CKD) are affected by enhanced oxidative stress and chronic inflammation [1,2], and both these conditions may contribute to increase the production of advanced glycation end products (AGE) [1,2]. Furthermore, serum AGE concentration is inversely related to residual renal function. Therefore, in these patients, AGE accumulation depends on both their increased production and reduced renal excretion [3,4,5]. In CKD patients, AGE accumulation may also accelerate the decline of renal function [3,6], and it is independently associated with all-cause mortality [3,4,6,7,8]. The detrimental effects of AGE are mainly due to the activation of the receptor for advanced glycation end products (RAGE). RAGE is a multiligand cell membrane receptor of the immunoglobulin superfamily, which, once activated, promotes production of intracellular reactive oxygen species (ROS), activation of nuclear factor-kB (NF-kB), and other intracellular signals. All these pathways regulate the synthesis of pro-inflammatory cytokines and other mediators that affect cell survival, differentiation, and proliferation, beyond inducing metabolic changes. RAGE is ubiquitously expressed at low levels on cell membranes, but, at increasing AGE levels, AGE-induced RAGE activation leads to a significant up-regulation of RAGE expression and amplification of the pro-inflammatory response [2,9]. Besides the cell-membrane form, RAGE also exists as a soluble molecule called sRAGE, which is composed by two different forms: the cleaved RAGE (cRAGE), which derives from proteolytic cleavage of the membrane-bound RAGE; and the endogenous secretory RAGE (esRAGE), a splice variant with the extracellular domains but without the intracytoplasmic and transmembrane domains [10]. sRAGE has been associated with an increased risk of cardiovascular and renal damage [11,12,13]. In the field of CKD, studies from our group suggested a role for sRAGE as a prognostic factor for mortality in end stage CKD patients under dialysis displaying an active process of cardiac remodelling [12]. Furthermore, among the different sRAGE forms, esRAGE and cRAGE/esRAGE ratio, along with AGE, were independently associated with all-cause mortality in advanced CKD patients [7]. However, since these molecules were quantified at a singular time point, we were not able to demonstrate whether and how they can vary spontaneously over time in CKD patients with an advanced impairment of renal function.

To fill this knowledge gap, in the current study, we aimed to quantify AGE, sRAGE, its different isoforms, cRAGE and esRAGE, and AGE/sRAGE ratio at baseline and follow-up, in a cohort of older patients affected by CKD. Secondly, we wanted to investigate whether changes in the levels of these parameters and the ratio of AGEs/sRAGE were associated with eGFR changes.

## 2. Materials and Methods

### 2.1. Patients and Study Design

We prospectively observed 64 prevalent patients that attended our outpatient CKD clinic between 09/2016 and 03/2018 with an eGFR <60 mL/min/1.73 m^2^. As prespecified in PROVE study protocol, we analyzed at baseline and 12 months those patients that were still on follow-up at our outpatients’ clinic at the end of the study. The causes of drop-out are depicted in Figure 1. Patients were selected according to the following criteria: age ≥65 years, CKD stages 3b to 5, and not yet on dialysis and relatively stable eGFR over the previous 6 months (with less than 2 mL/min/1.73/m^2^ of variation). eGFR was estimated according to the CKD-EPI formula. We applied some exclusion criteria: active and advanced cancer, decompensated chronic liver diseases (advanced cirrhosis and/or ascites), severe heart failure (NYHA class III–IV), nephrotic syndrome, hypothyroidism, hyperthyroidism, malabsorption diseases, and inability to cooperate. We also excluded all patients that were under treatment with immunosuppressive drugs or had a hospitalization in the previous three months. Twenty-four hours of urinary collection was started in the morning of the day preceding the visit, and biochemical parameters were sampled the morning of the visit after an overnight fast of at least 12 h. The study was conducted according to the ICP Good Clinical Practices Guidelines, and it was approved by the Ethics Committee of our Institution (Milano 2- approval N. 347/2010). All patients signed an informed consent.

### 2.2. sRAGE, esRAGE and cRAGE Quantification

Quantification of sRAGE and its soluble forms were performed as previously indicated [7]. Briefly, sRAGE was measured by an ELISA kit from R&D Systems (DY1145, Minneapolis, MN, USA). esRAGE was quantified by the ELISA kit from B-Bridged International (K1009–1, Santa Clara, CA, USA). The intra- and inter-assay coefficients of variation of esRAGE assay were 6.37 and 4.78–8.97%, respectively. We obtained cRAGE level by subtracting esRAGE from total sRAGE. The AGE/sRAGE ratio was then obtained. The GloMax^®^-Multi Microplate Multimode Reader was used for photometric measurements (Promega, Milan, Italy).

### 2.3. AGE Quantification

AGEs were quantified by measuring the fluorescence intensity of plasma samples at 414–445 nm after excitation at 365 nm, as previously reported [14,15], using a fluorescence spectrophotometer (The GloMax^®^, Promega, Milan, Italy). Fluorescence was expressed as the relative fluorescence intensity in arbitrary units (AU). AGEs were then normalized for total protein content.

### 2.4. Statistical Analysis

Continuous variables were expressed as mean with standard deviation (SD), in parametric distributions or median with interquartile range (IQR), in non-parametric data. Categorical variables were summarized as percentages. Variables comparison between baseline and follow-up were performed by paired t-test for parametric variables, while intra-group comparison of non-parametric ones were done by using Wilcoxon test.

Comparison between variations of AGES and RAGEs isoforms and eGFR were performed by GLMmodel.

Analysis was conducted using IBM SPSS software (version 25, IBM, Armonk, NY, USA).

## 3. Results

### 3.1. Patients’ Characteristics

Patients’ characteristics are depicted in Table 1. Median age was 81 years and male patients were preponderant (70%). Most of our patients were hypertensive (91%), and a considerable part were diabetic (63%). Pharmacological treatments at baseline and follow-up were reported in Appendix A. Baseline biochemical characteristics of the recruitable cohort vs. the one that was not eligible due to FU dropout are available at Appendix A.

### 3.2. Metabolic and Renal Function Parameters at Baseline and Follow-Up (FU)

Table 2 shows the evolution of the main metabolic and renal function variables from baseline to FU. Notably, only eGFR and fasting glucose levels showed a significant variation from baseline to FU. In particular, eGFR showed a slightly decreasing trend (27 ± 10 vs. 24 ± 9, *p* = 0.004), while fasting glucose levels increased at FU (105 [91;144] vs. 106 [91;142], *p* = 0.02). This rising pattern, even if not statistically significative, was shared by triglyceride and uric acid levels. We encountered no difference in HbA1c levels from baseline to FU. Finally, no difference was found between albumin levels, an indirect indicator of general nutritional status and inflammation and in Prot-U levels.

### 3.3. Changes of AGEs, sRAGE and Its Isoforms, and AGEs/sRAGE Ratio from Baseline to Follow-Up and Their Association with eGFR Variations

The distribution of AGEs, sRAGE, esRAGE, and cRAGE at baseline and FU are shown in Figure 2. sRAGE and its isoforms significantly decreased from baseline to FU (sRAGE BS vs. FU: 1679 [1393;2038] vs. 1442 [1117;2102], *p* = 0.009—esRAGE BS vs. FU: 508 ± 224 vs. 449 ± 193, *p* = 0.009—cRAGE BS vs. FU: 1173 [981;1557] vs. 974 [815;1499], *p* = 0.019). Conversely, AGEs concentrations were stable between baseline and FU values.

We also evaluated the association between the differences (basal–FU) of AGEs (ΔAGEs), sRAGE (ΔsRAGE), esRAGE (ΔesRAGE), cRAGE (ΔcRAGE), and eGFR (ΔeGFR) (Figure 3). No significant associations with ΔeGFR were found for any of the parameters evaluated. It can just be noted that ΔAGEs inversely correlated with ΔeGFR, even if not with a statistically significant association (R = 0.201, *p* = 0.118).

The distribution of AGEs/sRAGE ratio at baseline and follow-up were also investigated. In this case, we found a statistically significant difference, with higher values at FU compared to baseline, as shown in Figure 4A. Considering this result, we tested the association of ΔAGEs/sRAGE with ΔeGFR in our patients (Figure 4B). The trend of association previously noted for ΔAGEs and ΔeGFR in Figure 3 was observed also in this case. In fact, a rising of ΔAGEs/sRAGE ratio was partially associated with a decrease in ΔeGFR. However, also in this case, the observed associations were not statistically significant associations (R = 0.174, *p* = 0.176).

## 4. Discussion

The main findings of our study can be summarized as follows: (1) in CKD patients with a relative stability of eGFR, at 12 months follow-up, mean AGEs levels were almost the same, whereas sRAGE as well as the two specific isoforms, esRAGE and cRAGE, were reduced; (2) AGEs/sRAGE ratio was increased, instead; (3) variations of AGEs and RAGEs isoforms were not correlated with eGFR changes during follow-up.

To the best of our knowledge, most of the studies carried out up until today in CKD patients correlated AGE and/or sRAGE to eGFR and other clinical outcomes by performing a single point quantification of these variables. Furthermore, previous studies considered only AGEs and/or sRAGE. The novelty of our study deals with the evaluation of changes occurring in the levels of AGEs, sRAGE and its different forms, as well as the AGEs/sRAGE ratio at 1-year follow-up.

Diabetes mellitus, hypertension, and dyslipidemia are known to be strongly associated with co-morbidities in CKD patients and to increase the risk of mortality, mainly due to cardiovascular events. Diabetes mellitus (DM) is a recognized risk factor for the onset and progression of many disorders, among which cardiovascular diseases have a great impact on mortality [16]. The production of superoxide in mitochondria and AGEs synthesis contribute to increased oxidative stress in DM. In particular, AGEs by binding to RAGE, induce ROS production, amplifying RAGE expression and the inflammatory response [17]. Furthermore, AGEs increase the activity of renin-angiotensin system and stimulate the production of transforming growth factor-beta (TGF-β) which contribute to induce micro and macrovascular damage and induce a pro-fibrotic response [18].

Patients affected by DM have an increased risk to develop CKD too. In fact, in CKD patients, it was observed that the progressive reduction of eGFR determines an increased retention of AGEs that contributes to accelerate the progression of renal and cardiovascular damage, regardless of the presence of DM. CKD is related to cardiovascular disorders through different mechanisms which include: (1) the production of inflammatory mediators and ROS; (2) the accumulation of AGEs and many uremic toxins, among which indoxyl sulfate is associated with altered monocytes activation, intensified inflammatory process, and oxidative stress; (3) the toxicity of phosphate and the activation of FGF-23 pathway [19,20]. All these pathways are strongly related each other, and all contribute to induce diastolic disfunction, left ventricular hypertrophy, and increase the risk of heart failure [21]. Therefore, both DM and CKD may independently promote AGEs production and accumulation that may ultimately contribute to increasing the cardiovascular risk, which is the utmost in subjects with diabetic kidney disease (DKD) [16].

Recently, novel risk factors have emerged to be relevant in CKD, and the overproduction of ROS and/or a reduction in antioxidant defense capacity seem to play a pivotal role in CKD progression and CKD-related complications [22]. In CKD, significant upregulation of nicotinamide adenine dinucleotide phosphate oxidase and downregulation of superoxide dismutase increase the levels of superoxides [23], which in turn, by reacting with nitric oxide (NO), produce peroxynitrite and cause nitrosative stress. Hydrogen peroxide and chloride ions also play a role. After being metabolized by myeloperoxidase to hypochlorous acid, they contribute to chlorinated stress. Reduced NO production, hypertension, and angiotensin II activity might also contribute to ROS production in CKD [24,25]. A pro-oxidant state can occur as early as in CKD stage 3, as evidenced by increased levels of biomarkers related to oxidative stress [26].

Previous studies indicated that the pro-oxidant, inflammatory, and uremic environment of CKD and the reduced kidney filtration were among the main causes promoting AGEs synthesis and accumulation [27]. AGEs are recognized risk factors for the progression of both CKD and other co-morbidities affecting these patients, such as cardiovascular disorders, sarcopenia, and frailty [3,28,29,30,31,32]. Furthermore, we have recently shown that AGEs may help to stratify overall mortality risk in patients with advanced CKD [7], thus emphasizing the importance of AGEs as a biomarker for implementing the clinical evaluation of CKD patients to improve their overall outcomes. The observation that AGEs levels did not change at 1 year follow-up, although the slight decrease in renal function could suggest that the concentration of these products were more related to the individual pro-oxidant environment than to renal function. Unfortunately, we cannot confirm the lack of changes in patient’s oxidative stress, but the high levels of pro-inflammatory mediators observed at baseline in the same cohort of patients [33] seem to suggest the achievement of a plateau by those mediators that can promote AGEs synthesis.

AGE are toxic compounds, but their damaging effects depend not just on their levels, but also on the function of endogenous defensive mechanisms that can reduce their synthesis and block their detrimental effects. Among them, sRAGE is a powerful protective molecule. sRAGE, the soluble form of RAGE, is a decoy receptor that, by binding AGEs, reduces RAGE signaling at cellular levels and prevents RAGE-independent AGEs effects. The protective role of sRAGE has been shown in pre-clinical models in which sRAGE administration or blockage of membrane RAGE reduce different detrimental injuries triggered by RAGE stimulation, such as myocardial injury [34], neointimal expansion in arterial restenosis [35], and glomerulosclerosis [36]. Clinical observation confirmed a protective role of sRAGE and a decline of its levels in different diseases, such as coronary artery disease [37,38], sarcopenia [39], essential hypertension [40], chronic obstructive pulmonary disease [41], and hyperthyroidism [42]. Increased levels, instead, have been observed in diabetes mellitus and CKD, as an adaptive mechanism that tries to counteract the sharp increase in AGEs and reduced kidney function, respectively [43,44,45]. Therefore, the AGEs/sRAGE ratio seems to be a better marker to estimate patient’s risk than AGEs or sRAGE alone. The up regulation of AGEs/sRAGE ratio at follow-up suggests that the risk of AGE-dependent progression of both CKD and other co-morbidities were higher in these patients. An interesting result that we observed in our work was that there was a trend of inverse proportionality between AGEs and AGEs/RAGEs ratio variation and eGFR. The lack of significative association between changes in AGEs and AGEs/sRAGE ratio and variation of eGFR at 1-year follow up could be explained by the fact that the observation time may have been too short to appreciate a further decrease in kidney function due to the imbalance of the AGEs/RAGE system. However, this does not exclude that the rise in AGEs/sRAGE could be associated with the progression of other co-morbidities and factors that have not been evaluated in this study because out of our aim.

We did not explore the molecular mechanisms affecting sRAGE reduction, but we can confirm that both cRAGE and esRAGE contribute to this decline. Although the two forms derive from different mechanisms, we can hypothesize that are similarly affected in their synthesis or in their metabolism.

Considering that the imbalance in the levels of AGEs and sRAGE can play a role in CKD progression which, in turn, further promotes AGE accumulation, therapeutic strategies aimed to reduce AGEs synthesis as well as AGEs-related detrimental effect could find application in CKD, with the aim to slow the progression of the disease and of the other co-morbidities. Our study has several limitations. First of all, the small number of patients and the short follow up, and secondarily, that eGFR did show only a small variation at follow up. However, we adopted quite strict exclusion criteria in order to avoid other conditions that may have influenced AGEs and sRAGE variation.

Overall, the fact that eGFR was substantially stable allowed us to exclude those variations of AGEs and sRAGE that were merely determined by accumulation, secondary to eGFR worsening. The results highlighted in our study could be an initial hint of the true harming potential of AGEs in CKD patients. Since several molecules have been proposed as potential anti-AGEs therapy, among these, the use of sRAGE itself, our results may prompt design of future interventional strategies finalized to contrast AGEs effects in CKD patients.

## 5. Conclusions

In a cohort of patients with a stable clinical condition at 1 year follow-up, AGEs/sRAGE were associated with renal function. The lack of association with eGFR suggests that other factors can influence its increase. AGEs/sRAGE can therefore be an additional risk factor for CKD progression in a longer time, but its role as a prognostic tool needs further investigation.

## Figures and Tables

**Figure 1 antioxidants-10-01994-f001:**
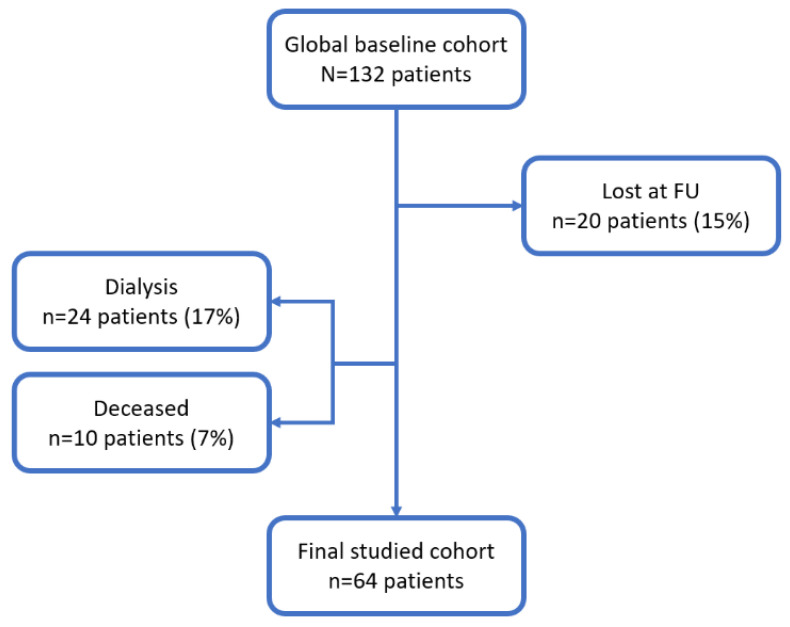
Population selection criteria.

**Figure 2 antioxidants-10-01994-f002:**
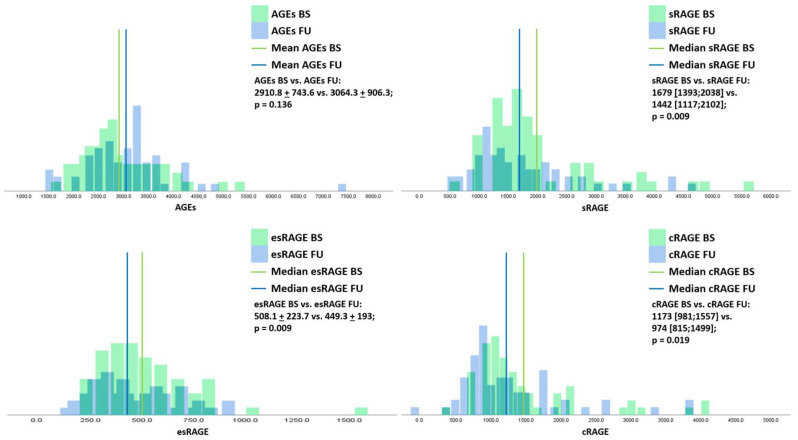
Distribution of advanced glycation end products (AGEs), and soluble receptor for AGE (sRAGE) and its isoforms’ distribution at baseline and follow-up. BS: baseline; FU: follow-up; AGEs: advanced glycation end products; sRAGE: soluble receptor for advanced glycation end products receptor; esRAGE: endogenous secretory receptor for AGE; cRAGE: clivated receptor for AGE.

**Figure 3 antioxidants-10-01994-f003:**
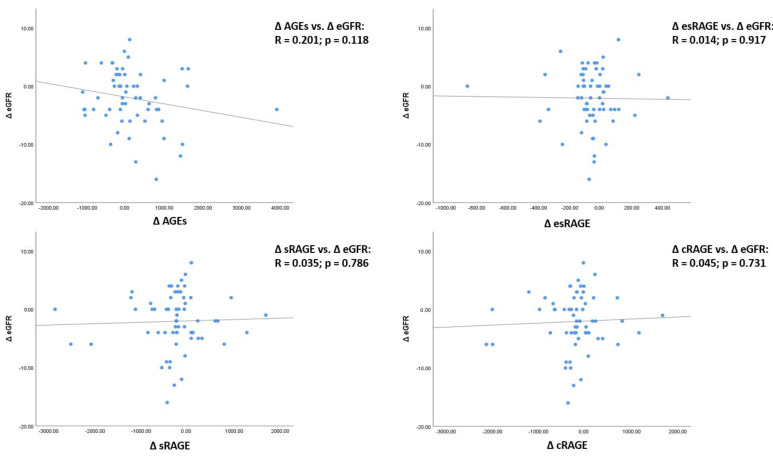
Linear regression analyses between the differences (basal–FU) of advanced glycation end products (ΔAGEs), soluble receptor for AGEs (ΔsRAGE) and its isoforms (ΔesRAGE and ΔcRAGE) and eGFR (ΔeGFR). Δ: difference; BS: baseline; FU: follow-up; AGEs: advanced glycation end products; sRAGE: soluble receptor for advanced glycation end products receptor; esRAGE: spliced soluble receptor for AGE; cRAGE: cleaved receptor for AGE.

**Figure 4 antioxidants-10-01994-f004:**
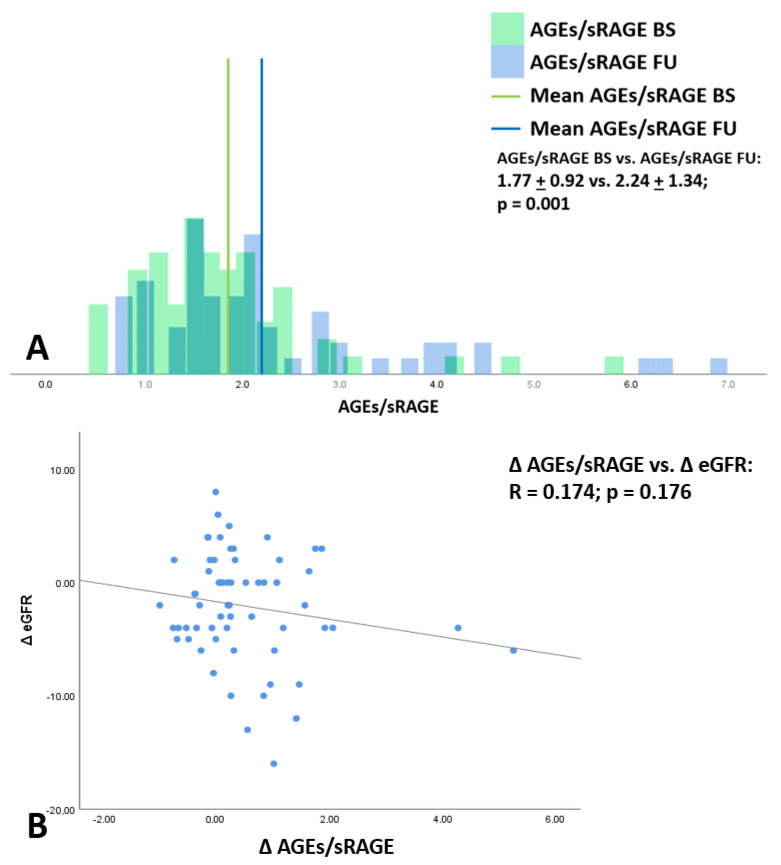
Distribution of advanced glycation end products/soluble receptor for advanced glycation end products ratio. AGEs/sRAGE at baseline and follow-up (panel (**A**)). Linear regression analysis between the differences (basal–FU) of ΔAGEs/ΔsRAGE and eGFR (ΔeGFR) (panel (**B**)). BS: baseline; FU: follow-up; AGEs: advanced glycation end products; sRAGE: soluble receptor for advanced glycation end products receptor; eGFR: estimated glomerular filtration rate.

**Table 1 antioxidants-10-01994-t001:** Patients’ cohort characteristics.

Variables	Overall Cohort (*n* = 64)
*General characteristics*	
Age, (years)	81 [75;85]
Males, *n* (%)	45 (70)
Diabetes, *n* (%)	40 (63)
Hypertension *n* (%)	58 (91)
eGFR, (mL/min/1.73 m^2^)	27 ± 10
Prot-U 24 h (g/24 h)	440 [201;1037]
BMI (kg/m^2^)	28.4 ± 4.6
Waist circumference, (cm)	98 [93;103]
Arterial systolic pressure (mmHg)	137 [127;145]
Arterial dyastolic pressure	78 [67;85]
*Metabolic characteristics*	
Uric Acid (mg/dL)	6.0 ± 1.4
HbA1c (mmol/dL)	49 ± 12
Fasting blood Glucose (mg/dL)	105 [91;144]
Total Cholesterol (mg/dL)	162 ± 27
HDL-Cholesterol (mg/dL)	47 [40;58]
LDL-Cholesterol (mg/dL)	81 [71;100]
Triglycerides (mg/dL)	120 [100;162]
Albumin (g/dL)	4 ± 0.4
*AGEs and RAGEs*	
AGEs (arbitrary unit)	3033 ± 789
sRAGE (pg/mL)	1341 [1453;2858]
esRAGE (pg/mL)	508 ± 224
cRAGE (pg/mL)	1173 [982;1557]
AGEs/sRAGE (arbitrary unit)	1.7 ± 0.9
*Lost at Follow-up (global cohort n = 132)*	
Dialysis *n* (%)	24 (17)
Death *n* (%)	10 (7)

Mean + SD (standard deviation) was used for continuous parametric data; median (IQR—interquartile range) was used for non-parametric data; number (*n*) and (%) were used for categorical variables. eGFR: estimated glomerular filtration rate; Prot-U: proteinuria in a 24 h sample; BMI: body mass index; HbA1c: glycated hemoglobin; HDL-Cholesterol: high density lipoprotein-Cholesterol; LDL-Cholesterol: low density lipoprotein-Cholesterol; AGEs: advanced glycation end products; sRAGE: soluble receptor for AGE; esRAGE: endogenous secretory receptor for AGE; cRAGE: cleaved receptor for AGE.

**Table 2 antioxidants-10-01994-t002:** Metabolic and renal function parameters at baseline and follow-up.

Variables	Overall Cohort Baseline (*n* = 64)	Overall Cohort Follow-Up (*n* = 64)	*p*
eGFR, (mL/min)	27 ± 10	24 ± 9	**0.004**
Prot-U 24 h (g/24 h)	440 [201;1037]	300 [166;882]	**0.288**
Arterial systolic pressure (mmHg)	138 [127;145]	140 [127;150]	**0.371**
Arterial dyastolic pressure (mmHg)	78 [67;85]	80 [70;85]	**0.942**
Fasting blood Glucose (mg/dL)	105 [91;144]	106 [91;142]	**0.02**
Albumin (g/dL)	4.1 ± 0.28	4 ± 0.29	0.28
Uric Acid (mg/dL)	6 ± 1.4	6.7 ± 7.9	0.41
Total Cholesterol (mg/dL)	162 ± 27	159 ± 31	0.47
HDL-Cholesterol (mg/dL)	47 [40;58]	45 [38;60]	0.26
LDL-Cholesterol (mg/dL)	81 [71;100]	81 [63;99]	0.18
Triglycerides (mg/dL)	120 [100;162]	133 [91;174]	0.09
HbA1c (mmol/dL)	49 ± 12	49 ± 12	0.70

Mean + SD (standard deviation) was used for continuous parametric data; median (IQR—interquartile range) was used for non-parametric data; *p* = 0.05 was the cut off for statistical significance. eGFR: estimated glomerular filtration rate; Prot-U: proteinuria in a 24 h sample; HDL-Cholesterol: high density lipoprotein-Cholesterol; LDL-Cholesterol: low density lipoprotein-Cholesterol; HbA1c: glycated hemoglobin.

## Data Availability

The data is contained within the article or Appendix A.

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
