# Peer review of "AGEs and sRAGE Variations at Different Timepoints in Patients with Chronic Kidney Disease"

_antioxidants, 2021, doi:10.3390/antiox10121994_

Round 1

Reviewer 1 Report

The study design is appropriate to the objective. Methods are clearly described and result are presented in tables and graphs easy to read.

I have a methological comment: as AGEs and sRAGE are measured in the same patients at different time points, statistical significance of results should be corrected for the effect of repeated meauser. A simple methods is to apply the Bonferroni correction for repeated measures, while a more solid method is to apply a GLM model.

Discussion is well conducted and biological palusibility of the observed relationship is deeply discussed

Reviewer 2 Report

Dear Authors,

Thank you for the chance to review your paper entitled AGEs and sRAGE variations at different timepoints in patients with chronic kidney disease.

The paper describes an important clinical matter - the interplay between CKD milieu, oxidative stress and inflammation in the sight of the RAGE. I found paper as an interesting approach and for sure stronger than basic science due to incorporating human/clinical data. 

Based on my feelings and knowledge I have some major issues that MUST be solved/answered before moving forward. 

Firstly, I have an impression of the "salami" publication here. I am thinking that you might already partially publish the data in your previous papers, namely: doi: 10.1186/s12877-020-01757-8 ; doi: 10.3390/toxins13070472 ; 

Please provide your comments and describe the potential overlapping of the data.

Moreover, I am so surprised by what I am seeing in Table 2 - comparing the glucose levels (that are so important for the paper) are almost the same, and that's the fact. Your P-value of 0.02 is impossible to achieve in these groups. eGFR values with SD look already worse. Please double-check. 

There is no information about the pharmacological treatment of the patients during the follow-up period. This data must be included otherwise the conclusions are so highly biased.

In fact, your results and conclusions stand against many other papers and even basic KDIGO datasheet facts. Can you point out any factors that might lead to this?

What was the percentage of the lost cohort? Taking into account the age of the patients it is a must-have incident during a 12-month follow-up period, sadly. 

Minors:

-no AGE/RAGE abbs explained in the abstract. Abstract must be readable when stands alone

-Figure 2 shows lots of outfitters. Please revise your inclusion criteria.

-no control values/control group was given. 

-no proper discussion explaining the general levels of the ox products.

Reviewer 3 Report

This is an interesting study, in which possible correlations between levels of molecules related to AGEs and eGFR are analyzed in patients with chronic kidney disease, during a follow-up of 1 year.

In my opinion, it is a very preliminary study, as indicated by the authors at the end of the discussion. Despite this, it could improve in the following aspects:

- The description of the patients should be completed with the treatment guidelines followed, as well as their possible variation during the year of follow-up. Likewise, analyze the possible statistical correlation between these treatment factors and the variables related to the determined AGEs.

- Another aspect that I find interesting is the need to provide the evolution of other parameters throughout the year of follow-up, such as blood pressure, for example.

Finally, the relationship between AGEs, oxidative stress, and inflammatory mediators should be indicated by some biochemical variables, their evolution during the year of follow-up analyzed, and an attempt to relate these values to the molecules related to AGEs and eGFR.

Round 2

Reviewer 2 Report

The Authors improved the manuscript significantly as well as added supplementary files that include important data. I am still slightly confused with the p value of 0.02 for Fasting blood Glucose (mg/dl) but I trust the Authors in this matter.

I think that the only one missing thing right now is introducing few senetnces regarding the possbility of the interplay between oxidative stress and CVD in CKD - the Authors should explain this section since CVD links oxidation, diabetic status as well as CKD - so this is the cross-section of DKD. Below pasted papers might be handy: 

https://doi.org/10.1186/s12882-017-0457-1

doi: 10.2337/dci17-0053

https://doi.org/10.1161/CIRCULATIONAHA.120.051898

After this change is implemented, I am supportive for the publication of the paper.

Reviewer 3 Report

.

Author Response

No comments available in "Comments and suggestions". The only comments and revisions were given by Reviewer 2, and we have revised our manuscript accordingly. Hope these changes fit all your requests.

Thank you for your time

Best regards

Simone Vettoretti